



# Meeting climate targets by direct $CO_2$ injections:
# What price would the ocean have to pay?

Fabian Reith[1], Wolfgang Koeve[1], David P. Keller[1], Julia Getzlaff[1], and Andreas Oschlies[1]

[1]GEOMAR Helmholtz-Centre for Ocean Research Kiel, Düsternbrooker Weg 20, 24105 Kiel, Germany

*Correspondence to*: Fabian Reith (FReith@geomar.de)

**Abstract.** We investigate the climate mitigation potential and collateral effects of direct injections of captured $CO_2$ into the deep ocean as a possible means to close the gap between an intermediate $CO_2$ emissions scenario and a specific temperature target, such as the 1.5°C target aimed for by the Paris Agreement. For that purpose, a suite of approaches for controlling the

amount of direct $CO_2$ injections at 3000m water depth are implemented in an Earth System Model of intermediate complexity.

Following the representative concentration pathway RCP4.5, which is a medium mitigation $CO_2$ emissions scenario, cumulative $CO_2$ injections required to meet the 1.5°C climate goal are found to be 390 Gt C by the year 2100 and 1562 Gt C at the end of simulations, by the year 3020. The latter includes a cumulative leakage of 602 Gt C that needs to be re-injected in

order to sustain the targeted global mean temperature.

$CaCO_3$ sediment and weathering feedbacks reduce the required $CO_2$ injections that comply with the 1.5°C target by about 13 % in 2100 and by about 11 % at the end of the simulation.

With respect to the injection-related impacts we find that average pH values in the surface ocean are increased by about 0.13 to 0.18 units, when compared to the control run. In the model, this results in significant increases in potential coral reef habi-

tats, i.e., the volume of the global upper ocean (0 to 130m depth) with omega aragonite > 3.4 and ocean temperatures between 21°C and 28°C, compared to the control run. The potential benefits in the upper ocean come at the expense of strongly acidified water masses at depth, with maximum pH reductions of about -2.37 units, relative to preindustrial, in the vicinity of the injection sites. Overall, this study demonstrates that massive amounts of $CO_2$ would need to be injected into the deep ocean in order to reach and maintain the 1.5°C climate target in a medium mitigation scenario on a millennium timescale,

and that there is a trade-off between injection-related reductions in atmospheric $CO_2$ levels accompanied by reduced upper-ocean acidification and adverse effects on deep ocean chemistry, particularly near the injection sites.

## 1. Introduction

The Paris Agreement of December 2015 has set the political target of limiting global warming to well below 2°C, if not 1.5°C, above preindustrial levels (UNFCCC, 2015). Staying within the Paris target range is perceived as a safe limit that

avoids dangerous anthropogenic climate change and ensures sustainable food production and economic development (Rock-



ström et al., 2009; Knutti et al., 2015; Rogelj et al., 2016). As a first step towards meeting the Paris climate goals, countries have outlined national post-2020 climate action plans by submitting their Nationally Determined Contributions (NDCs) to climate mitigation in order to meet the <2°C climate target (e.g., Clémençon, 2016). However, even if these NDCs are fully realized, it is estimated that a median warming of 2.6 to 3.1°C will occur by the year 2100 (Rogelj et al., 2016). Consequent-

ly, it is questionable whether conventional measures currently considered by individual states will be sufficient to reach and maintain the <2°C climate target (e.g., Horton et al., 2016).

The scientific rationale of such claims is based on observational records and results of climate models of varying complexity that have found a tight correlation between cumulative $CO_2$ emissions and global mean temperature (Allen et al., 2009; Matthews et al., 2009; MacDougall, 2016). From this transient climate response to cumulative carbon emissions (TCRE) it can

be estimated that the total quota of $CO_2$ emissions from all sources (fossil fuel combustion, industrial processes and land-use change) that is compatible with a 1.5°C target will be used up in a few years at current emission rates (Knopf et al., 2017; Mengis et al., 2018), and for a 2°C target it is likely to be reached in the next 2 to 3 decades (Friedlingstein et al., 2014). Thus, the window of opportunity for deep and rapid decarbonization that would allow for such a climate target through emissions reduction alone is closing soon (Sanderson et al., 2016).

45        Given the very challenging and urgent nature of the task of reaching the agreed-upon Paris climate goals, unconventional methods are being discussed. Under specific consideration are negative emission technologies, i.e., measures that deliberately remove $CO_2$ from the atmosphere (e.g., Gasser et al., 2015) and store it somewhere else, e.g., in geological reservoirs or the deep ocean (e.g., IPCC, 2005). Negative emissions are already included in all realistic scenarios from integrated assessment models (IAMs) that limit global warming to <2°C or less above preindustrial levels (Collins et al., 2013;

Rockström et al., 2016; Rogelj et al., 2016). However, none of the currently debated negative emissions technologies, such as bioenergy with carbon capture and storage (BECCS), direct air capture with carbon storage (DACCS) and enhanced weathering (EW), appears to have, regardless of the scenario, the potential to meet the <2°C target without significant impacts on land, energy, water or nutrient resources (Fuss et al., 2014; Smith et al., 2016; Williamson, 2016; Boysen et al., 2017).

55        One other option that has been considered is ocean carbon sequestration by direct injection of $CO_2$ into the deep ocean (e.g., Marchetti, 1977; Hoffert et al., 1979; Orr et al, 2001; Orr, 2004; IPCC, 2005; Reith et al., 2016). The $CO_2$ could be derived from point sources such as power plants or direct air capture facilities, and thereby contribute to the carbon sequestration part of CCS, DACCS or BECCS. Direct injection of $CO_2$ into the deep ocean can also be thought of as the deliberate acceleration of the oceanic uptake of atmospheric $CO_2$, which happens naturally via invasion and dissolution of $CO_2$

into the surface waters, albeit at a relatively slow rate limited by the sluggish ocean overturning circulation. On millennial timescales, about 65-80% of anthropogenic $CO_2$ is thought to be taken up by the ocean via gas exchange at the ocean surface and by entrainment of surface waters into the deep ocean. This portion rises to 73-93% on timescales of tens to hundreds of



millennia via neutralization of carbonic acid with sedimentary calcium carbonate ($CaCO_3$) (e.g., Archer et al., 2005; Zeebe, 2012). Directly injecting $CO_2$ into the deep ocean could speed up this natural process by directly accessing deep waters,

some of which remain isolated from the atmosphere for hundreds or thousands of years (DeVries and Primeau, 2011; their Figure 12), and by bringing the anthropogenic $CO_2$ in closer contact with the sediment where carbonate compensation reactions occur. This would prevent anthropogenic $CO_2$ from having an effect on the climate in the near future, and accelerate eventual and nearly permanent removal via reaction with $CaCO_3$ sediments.

Despite the well-known potential of the ocean to take up and store carbon (e.g., Sarmiento and Toggweiler, 1984;

Volk and Hoffert, 1985; Sabine et al., 2004;), direct $CO_2$ injection into the deep ocean is currently not allowed by the London Protocol and the Convention for the Protection of the Marine Environment of the North East Atlantic (OSPAR Convention) (Leung et al., 2014). A main concern that led to the current ban is that direct $CO_2$ injection will harm marine ecosystems in the deep sea, e.g., cold-water corals and sponge communities, at least close to the injection site (e.g., IPCC, 2005; Schubert et al., 2006; Gehlen et al., 2014). As emphasized by Keeling (2009) and Ridgwell et al. (2011) there are, however,

trade-offs between injection-related damages in the deep ocean and benefits at the ocean surface via a reduction in atmospheric $pCO_2$ and a decrease in upper ocean acidification. These should be discussed in relation to other mitigation options, that probably all imply offsetting some local harm against global benefits. Our current study aims to inform such a debate by providing quantitative information about impacts on ocean carbonate chemistry caused by direct injection of $CO_2$ into the deep ocean as a potential measure to reach and maintain a specific temperature target as given by the Paris climate targets.

For this purpose, we consider direct injection of $CO_2$ into the deep ocean as 'oceanic CCS', deposing $CO_2$ from point sources such as fossil fuel or biomass-based power plants or direct air capture plants. We assume that aggressive emissions reduction has led from a business-as-usual $CO_2$ emission scenario to a world with intermediate $CO_2$ emissions such as the one represented by the Representative Concentration Pathway (RCP) 4.5. Model-predicted global mean surface air temperatures for the RCP 4.5 $CO_2$ emission scenario range between 1.7°C and 3.2°C in the year 2100 (Clarke et al., 2014),

which is approximately in agreement with the warming after full achievement of current NDCs. Consequently, the 1.5°C climate target would not be reached under the RCP 4.5 scenario and is likely to be exceeded after the year 2050 (IPCC, 2014). We here explore the potential as well as collateral oceanic effects of 'oceanic CCS' as a means to fill the gap between emissions and climate impacts of the RCP 4.5 and the 1.5°C target of the Paris agreement. Due to the fact that we neglect the effects of non-$CO_2$ forcing agents in our injection experiments, our results provide a lower limit estimate, i.e. the cumulative

$CO_2$ amount that would need to be at least injected into the deep ocean in order to comply with the desired target.

The paper is organized as follows: In section 2 we address the methodological framework by describing the UVic model and the experimental setup of our experiments. In section 3 the results and the discussion of our model simulations are presented. Section 4 outlines the conclusions.



## 95    2. Methods

### 2.1 Model description

The model used is version 2.9 of the University of Victoria Earth System Climate Model (UVic ESCM). It consists

of three dynamically coupled main components: a three-dimensional general circulation ocean model based on the Modular

Ocean Model MOM2 (Pacanowski, 1996) including a marine biogeochemical model (Keller et al., 2012), a dynamic-

thermodynamic sea-ice model (Bitz and Lipscomb, 1999) and a $CaCO_3$-sediment model (Archer, 1996). The UVic ESCM

further includes a terrestrial vegetation and carbon-cycle model (Meissner et al., 2003) based on the Hadley Center model

TRIFFID (Top-down Representation of Interactive Foliage and Flora Including Dynamics) and the hydrological land com-

ponent MOSES (Met Office Surface Exchange Scheme), and a one-layer atmospheric energy-moisture balance model (based

on Fanning and Weaver, 1996). All components have a common horizontal resolution of 3.6° longitude x 1.8° latitude. The

oceanic component has 19 vertical levels with thicknesses ranging from 50 m near the surface to 500 m in the deep ocean.

Formulations of the air-sea gas exchange and seawater carbonate chemistry are based on the OCMIP abiotic protocol (Orr et

al., 1999). Marine sediment processes of $CaCO_3$ burial and dissolution are simulated using a model of deep ocean sediment

respiration (Archer, 1996).

### 2.2 Experimental design

For our default control run and injection experiments, the model has been spun up for 10,000 years under preindus-

trial atmospheric and astronomic boundary conditions and run from 1765 to 2005 using historical fossil-fuel and land-use

carbon emissions (Keller et al., 2014). From the year 2006 onwards simulations are forced with $CO_2$ emissions according to

the RCP 4.5 and the Extended Concentration Pathway (ECP) 4.5, which runs until the year 2500 (Meinshausen et al., 2011).

This forcing includes $CO_2$-emissions from fossil fuel burning as well as land-use carbon emissions, e.g. from deforestation.

After the year 2500, $CO_2$ emissions are assumed to decrease linearly until they cease at the end of the simulations in year

3020. In the default control run and injection experiments we do not apply greenhouse gas emissions other than $CO_2$, nor do

we simulate the effect of sulfate aerosols or non-$CO_2$ effects of land use change. Further, prescribed monthly varying winds

from the National Center for Environmental Prediction (NCEP) reanalysis are used together with a dynamical feedback from

a first-order approximation of geostrophic wind anomalies associated with changing winds in a changing climate (Weaver et

al., 2001).

Simulated $CO_2$ injections are based on the OCMIP carbon sequestration protocols (see Orr et al., 2001; Orr, 2004)

and carried out in an idealized manner by adding $CO_2$ directly to the dissolved inorganic carbon (DIC) pool, thus neglecting

any gravitational effects and assuming that the injected $CO_2$ instantaneously dissolves into seawater and is transported quick-

ly away from the injection point and distributed homogenously over the entire model grid box with lateral dimensions of a



few hundred kilometers and many tens of meters in the vertical direction (Reith et al., 2016). Consequently, the formation of

CO$_2$ plumes or lakes as well as the potential risk of fast rising CO$_2$ bubbles are neglected (IPCC, 2005; Bigalke et al., 2008).

The physical transport of the injected CO$_2$ and its transport pathways from the individual injection sites towards the

surface of the ocean are tracked by means of inert 'dye' tracers (one per injection site). At the injections sites, these tracers

are loaded at rates proportional to the amount of CO$_2$ injected. At the sea surface the tracers are subject to a loss to the at-

mosphere, which is computed in proportionality to the total CO$_2$ gas exchange and fractional contribution to total DIC of the

respective tracer at the ocean surface. The sum of tracer loss to the atmosphere from the individual 'dye' tracers provides an

estimate of the loss of injected carbon to the atmosphere.

Following Orr et al. (2001), Orr (2004) and Reith et al. (2016) CO$_2$ is injected at seven separate injection sites,

which are defined as individual grid boxes near the Bay of Biscay (42.3°N, 16.2°W), New York (36.9°N, 66.6°W), Rio de

Janeiro (27.9°S, 37.8°W), San Francisco (31.5°N, 131.4°W), Tokyo (33.3°N, 142.2°E), Jakarta (11.7°S, 102.6°E) and Mum-

bai (13.5°N, 63°E) (Reith et al., 2016; their Figure 1). Injected CO$_2$ is distributed equally among the seven injection sites.

Direct CO$_2$ injections are carried out in the vertical grid box ranging from 2580 to 2990 m water depth (hereafter referred to

as injection at 3000 m). Compared to shallower injection, this reduces leakage and increases retention time (e.g., Orr et al.,

2001; Orr, 2004; Jain and Cao, 2005; Ridgwell et al., 2011; Reith et al., 2016). At this depth, liquid CO$_2$ is denser than sea-

water, which has the additional advantage that any undissolved droplets would sink to the bottom rather than rise to the sur-

face.

### 2.3 Model experiments

Three conceptually different approaches for applying oceanic CCS are simulated using the UVic model: The first

approach (*A1*) assumes that all anthropogenic CO$_2$ emissions are injected after a warming of 1.5°C is realized for the first

time, the second approach (*A2*) injects, in every year, an amount of CO$_2$ that ensures that temperatures do not rise beyond the

1.5°C target, and the third approach (*A3*) injects an amount of CO$_2$ to ensure that atmospheric CO$_2$ concentrations follow the

RCP/ECP 2.6 scenario as closely as possible. All idealized approaches are designed to counter the excessive emissions of the

RCP 4.5 scenario by direct CO$_2$ injections into the deep ocean to reach and maintain a specific temperature target as given

by the 1.5°C target until the end of this century and for another millennium. Injections in A2 (A3) are interrupted when the

simulated annual mean surface air temperature (atmospheric pCO$_2$) falls below the respective climate target. We further

study how the simulation of CaCO$_3$ sediment feedbacks and associated continental weathering modifies required CO$_2$-

injections and its impacts on ocean biogeochemistry. Table 1 provides an overview of all conducted simulations and their

set-up from the year 2006 onwards.

In the first approach (*A1*), all further CO$_2$ emissions of the RCP 4.5 scenario are completely re-directed to the injection sites

after the global-mean surface air temperature has for the first time exceeded the 1.5°C target. Some committed warming



(e.g., Matthews and Caldeira, 2008; Gillet et al., 2011) occurs in these simulations due to past emissions and climate cycle

feedbacks. This committed warming is at some point overlaid by leakage of injected $CO_2$ out of the ocean (e.g., Orr, 2004;

Reith et al., 2016), as well as by oceanic and terrestrial carbon cycle feedbacks that lead to a $CO_2$ increase in the atmosphere

and respective additional warming (see section 3.1). To diagnose the contribution from leakage, we design a leakage-free

sensitivity simulation (*A1_ Comitw*), in which $CO_2$ emissions are set to zero once the 1.5°C target is reached, and no $CO_2$ is

injected into the deep ocean.

In contrast to the first approach, the second one (*A2)* keeps the global mean temperature at the defined threshold of

1.5°C, relative to preindustrial, by injecting as much $CO_2$ into the deep ocean as is necessary to maintain an annual mean

temperature that is only 1.5°C above preindustrial levels. We diagnose this amount of $CO_2$ using the transient response to

emissions (TCRE, Allen et al., 2009; Matthews et al., 2009; MacDougall, 2016) of our model and the difference of the mod-

eled annual mean atmospheric temperature and the target temperature. $CO_2$ is only injected if the modeled temperature is

above the target temperature. In order to avoid interference with seasonal and longer periodic fluctuations of atmospheric

temperature (sensitivity experiments, not shown) we apply a running-mean averaging time scale of 1000 days. $CO_2$ injection

rates required to reach the respective target are updated every 5 days, which is when the atmospheric and oceanic model

components are coupled. The rate of $CO_2$ injection taken out of the atmosphere can be larger than the actual $CO_2$-emissions

(RCP/ECP 4.5), i.e. constitute net negative $CO_2$ emissions.

In the third approach (*A3*), we inject the amount of $CO_2$ that is needed to follow the atmospheric $CO_2$ concentra-

tions of the extended Representative Concentration Pathway RCP 2.6 and Extended Concentration Pathway ECP 2.6, which

is a reference scenario that has been suggested to reach the <2°C climate target with a ≥66% probability (IPCC, 2014). From

year 2500 onwards, the targeted atmospheric $CO_2$ concentration is held constant until the end of the simulations. Therefore,

the model computes at every atmospheric time step the difference between its current simulated atmospheric $CO_2$ concentra-

tion, given the RCP 4.5 $CO_2$-emissions, and the targeted atmospheric $CO_2$ concentration from the RCP 2.6 pathway. This

difference is used to diagnose the $CO_2$-injection needed to keep the model's atmospheric $CO_2$ concentrations as close as pos-

sible to the RCP 2.6 concentration pathway. A respective amount of $CO_2$ is injected and subtracted from the prescribed $CO_2$

emissions to the atmosphere, which eventually results in net negative emissions. We apply temporal averaging and update

the required $CO_2$-injection every 5 days.

In sensitivity experiments (Table 1) we further investigate the effect of $CaCO_3$ sediment feedbacks and continental

weathering on the cumulative $CO_2$ injections and on seawater carbonate chemistry for the different approaches. The effect of

$CaCO_3$ sediment dissolution is thought to be relevant as $CO_2$ injected at depth may react relatively directly with sedimentary

$CaCO_3$ and increase $CaCO_3$ dissolution near or downstream of the injection sites, resulting in an accelerated neutralization of

this anthropogenic $CO_2$ compared to a situation where $CO_2$ slowly invades the ocean via air-sea gas exchange (Archer et al.,

1998; IPCC, 2005). Therefore, we investigate the effect of $CaCO_3$ sediment feedbacks in our simulations by running the



model with and without a sediment sub-model. The global average percent of $CaCO_3$ in sediments in our "*sed*" simulations (section 3.2) is about 31 % in the year 2020 and compares well to about 34.5 % derived from observations as reported in Eby

et al. (2009). To ensure that, in steady state (i.e., during the model spin-up), DIC and alkalinity are conserved, the UVic model with sediment module also has a simple representation of continental weathering to compensate the burial-related loss of DIC and alkalinity. From the model spin-up we diagnose the global terrestrial weathering flux of DIC as 0.12 Gt C yr$^{-1}$, and an alkalinity flux of 0.02 Pmol yr$^{-1}$. During the transient runs with sediment module, this weathering flux is held constant, whereas sedimentary $CaCO_3$ accumulation or dissolution is allowed to evolve freely. Consequently, ocean alkalinity

and DIC adjust in response to interactions between seawater, injected $CO_2$, and sediments. Simulations with the sediment/weathering sub-model are based on a separate set of spin-up experiments (50 000 years), drift runs and historical simulation that all employ the sediment/weathering sub-model. Hereafter, simulations performed with the sediment/weathering model are referred to by the subscript "*sed*" (Table 1).

Relevant carbonate system parameters that are not computed at model run-time are derived offline for all simulations by

means of the Matlab-version of *CO2SYS* (Lewis and Wallace, 1998; van Heuven et al., 2009; Koeve and Oschlies, 2012; currently available from http://cdiac.ess-dive.lbl.gov/ftp/co2sys), using carbonic acid dissociation constants of Mehrbach et al. (1973), as refitted by Dickson and Millero (1987), and other related thermodynamic constants (Millero, 1995).

## 3. Results and Discussion

### 3.1 Oceanic CCS and the 1.5°C climate target

Here, we present the cumulative mass of $CO_2$ injected in the default runs (without $CaCO_3$ sediments) of the different approaches and show how effective these are in reaching and maintaining the 1.5°C climate target.

In the default simulation of the first approach (*A1*) oceanic CCS starts in the year 2045 after the 1.5°C climate target has been exceeded for the first time at a corresponding atmospheric $CO_2$ concentration of about 466 ppmv (Figs. 1 a, b, c). Between years 2020 and 2045, about 278 Gt C have been emitted in form of $CO_2$ to the atmosphere, i.e. a small fraction of

the 1242 Gt C of total emissions corresponding the extended RCP 4.5 scenario between years 2020 and 3020. From 2045 until year 3020, all $CO_2$ emissions (964 Gt C in total) are directly injected into the deep ocean (Fig. 1 a), resulting in zero anthropogenic $CO_2$ emissions into the atmosphere for the remaining simulation. After injection starts in year 2045, the atmospheric $CO_2$ concentration decreases, but only until the year 2341, when a minimum of about 409 ppmv is reached (Fig. 1 c). The increase of atmospheric $CO_2$ from year 2342 onwards is a result of leakage of $CO_2$ injected into the deep ocean earli-

er. By the end of the simulation, a total amount of 437 Gt C has leaked back into the atmosphere (Fig. 1 d). Thus, only about 55 % of the total mass injected (964 Gt C) remains in the ocean until year 3020. From 2078 onwards, the land perennially turns into a carbon source with a total carbon loss of about 21 Gt C to the atmosphere.



Global mean temperature, relative to preindustrial, oscillates around the 1.5°C climate target within ± 0.02°C after injections started until the year 2200. Until then, this approach (*A1*) is thus nearly successful in reaching and maintaining the

1.5°C climate target. Subsequently, however, global mean surface air temperature shows a slow increase of up to 0.02°C until 2341 although atmospheric $CO_2$ still decreases. This warming signal is owed to the lagged response of the deep ocean to previously increasing atmospheric $CO_2$, i.e. committed warming, resulting in a decline of the ocean heat uptake from the atmosphere and thus in an increase in the global mean temperature (Zickfeldt and Herrington, 2015; Zickfeldt et al., 2016). In this simulation (*A1*), this feedback mechanism (see also Fig. S1) is overlaid by increasing leakage of injected $CO_2$ back

into the atmosphere, which becomes the dominating process for atmospheric warming as obvious from the atmospheric $CO_2$ increase after year 2342 (Figs. 1 c, d). Hence, the global mean air temperature shows a steeper increase until it reaches a maximum of about +2.2°C above preindustrial level at the end of the simulation (Fig. 1 b). Thus, on a millennial time scale, the *A1* simulation overshoots the 1.5°C climate target by about 0.7°C. By subtracting this diagnosed leakage of 437 Gt C from the cumulative $CO_2$ injections (964 Gt C), we determine the required $CO_2$ emission reduction (527 Gt C) (Fig. 1 e)

relative to the RCP/ECP 4.5 scenario to comply with a global mean temperature of about +2.2°C, relative to preindustrial, on a thousand-year timescale.

Oceanic CCS in the second approach (*A2*) starts as well in 2045 (Figs. 1 a, c). Global mean temperature oscillates around the 1.5°C climate target until year 2300 (Fig. 1 b). These oscillations get smaller over time until global mean temperature essentially stays at 1.5°C until the end of the simulation. We find that the oscillations arise in the applied model from

climate-sea-ice feedbacks under the near-term 1.5°C conditions (see Fig. S2). The terrestrial biosphere turns into a carbon source in 2061 and land-atmosphere carbon fluxes oscillate around zero until the end of the simulation. The total carbon loss from land to the atmosphere is about 75 Gt C. Atmospheric $CO_2$ concentrations show a continuous decline when global mean temperature is held at the aspired climate target (Figs. 1 b, c). This is caused by a decline in ocean heat uptake as mentioned above and consistent to an additional accumulation of heat in the atmosphere at constant atmospheric $CO_2$ concentra-

tions (e.g., Zickfeldt and Herrington, 2015; Zickfeldt et al., 2016), and which, in our second approach, needs to be counteracted by further $CO_2$ injections into the deep ocean.

By the end of the *A2* run, cumulative $CO_2$ injections amount to about 1562 Gt C, which is about 600 Gt C (62 %) higher than in the *A1* simulation. This amount of additional $CO_2$-injections is needed in order to reduce global mean warming at the end of the thousand-year simulation from 2.2°C in run *A1* to 1.5°C in run *A2*. In the *A2* run, the diagnosed mass of

injected $CO_2$ that has leaked into the atmosphere and has been reinjected into the deep ocean during the entire simulation adds up to about 607 Gt C until year 3020 (Fig. 1 d). Hence, about 61 % of the total mass injected (1562 Gt C) stays in the ocean. This results in a required $CO_2$ emission reduction of about 955 Gt C (Fig. 1 e), i.e., the amount of emission reduction necessary to comply with the 1.5°C climate target on a 1000-year timescale.





In the third approach, *A3*, oceanic CCS starts in year 2031 (Fig. 1 a) as the atmospheric $CO_2$ concentration caused by the RCP 4.5 $CO_2$ emission scenario starts to exceed the targeted RCP 2.6 atmospheric $CO_2$ concentration. Relative to preindustrial, global mean temperature continues to increase to a maximum of approximately +1.5°C in the year 2078 at a corresponding atmospheric $CO_2$ concentration of 433 ppmv (Figs. 1 b, c). Subsequently, temperature decreases until it reaches about +0.9 °C relative to preindustrial temperature, while the atmospheric $CO_2$ concentration reaches 327 ppmv at the end of the simulation (Figs. 1 b, c). Up to that point in time, cumulative $CO_2$ injections in the *A3* simulation amount to about 2200 Gt C (Fig. 1 a). In response to negative emissions the land turns into an atmospheric carbon source (Keller et al., 2018) between year 2076 and 2600 with a total loss of about 144 Gt C to the atmosphere. From year 2600 onwards, the carbon flux between the atmosphere and land is nearly zero (below 0.03 Gt C/yr$^{-1}$). By the end of the simulation, the diagnosed leakage of injected carbon adds up to about 900 Gt C (Fig. 1 d), which means that about 59 % of the injected $CO_2$ that remains in the ocean until year 3020. The required emission reduction to move from an RCP4.5 pathway to RCP 2.6 in the *A3* run is about 1300 Gt C (Fig. 1 e).

By the end of the *A3* simulation, cumulative $CO_2$ injections are about 636 Gt C (29 %) higher than in the *A2* simulation. This is also reflected in the higher diagnosed leakage by about 293 Gt C in total, when compared to the *A2* simulation. In the attempt to follow the atmospheric $CO_2$ concentration of the RCP2.6 (section 2.3), cumulative $CO_2$ injections are almost twice the amount of the cumulative $CO_2$ emissions difference between the RCP4.5 scenario and the RCP2.6 scenario applied here. This can be explained by the fact that deep oceanic CCS steepens the surface to deep DIC-gradient (Fig. S3 a) fostering a back transport to the surface ocean. Most of this enhanced deep water DIC is transported with the meridional overturning circulation to the Southern Ocean (south of 40°S), where the largest fraction of the total leakage occurs in our injection experiments (Fig. S3 b). By the end of the *A1* simulation, we find that about 60 % of the diagnosed leakage has outgassed in the Southern Ocean compared to about 77 % in *A2* run and about 80 % in the *A3* simulation. Overall, we find that, the higher the direct $CO_2$ injections into the deep ocean are, the higher the leakage (Figs. 1 a, d) and the higher the relative portion outgassed in the Southern Ocean.

What this means in terms of the effectiveness of oceanic CCS is further highlighted by the comparison of the required cumulative $CO_2$ injections of the three different approaches (*A1, A2,* and *A3*) and the respective required emission reductions needed to reach the run's specific climate target under a RCP/ECP 4.5 $CO_2$ emission scenario. As illustrated in Figs. 2 a, b, c, the approaches *A1, A2* and *A3* represent increasingly stringent climate targets as evident from decreasing atmospheric warming relative to preindustrial conditions. Cumulative $CO_2$ injections by the year 2100 are largely equivalent to the required emission reduction, because only a tiny fraction of injected $CO_2$ has outgassed until that point in time (Figs. 1 d, 2 a). However, by the end of the millennial injection experiments, cumulative $CO_2$ injections are much larger than the required emission reductions in year 3020 as indicated by the slopes of the eye-fitted lines in Figs. 2 b, c. This is due to the fact that the leakage in the injection experiments (Fig. 1 d) requires a larger $CO_2$ removal effort, i.e., $CO_2$ that leaks out has to be



reinjected. If there were no leakage of injected carbon, i.e., perfect storage, then the cumulative $CO_2$ injections would equal the required emission reductions.

### 3.2 Sensitivities to $CaCO_3$ sediment feedbacks and weathering fluxes

As illustrated in Fig. 2 b, cumulative $CO_2$ injections in the $A2_{sed}$ simulation are about 165 Gt C (11 %) smaller until
the year 3020 when compared to the $A2$ run (1562 Gt C). This smaller $CO_2$ injection is a result of two processes ($CaCO_3$ sediment dissolution and constant terrestrial weathering), which both have the net effect of adding alkalinity to the model ocean, when compared to the standard experiments without sediment feedbacks and continuous weathering fluxes. By the end of the simulation, average ocean alkalinity has increased by 32 mmol/m$^3$ in the $A2_{sed}$ run compared to an average value of 2422 mmol/m$^3$ in the $A2$ run. About 84 % of this increase in global mean alkalinity can be attributed to ocean CCS and
resulting sediment dissolution at depth, the rest is from ocean acidification-induced $CaCO_3$ dissolution according to the RCP 4.5 $CO_2$-emission scenario, as evident from the control run with sediment and weathering feedback. An increase in ocean alkalinity may enhance the oceanic uptake of atmospheric $CO_2$, however, only if waters with increased alkalinity arrive at the surface waters and lower surface-ocean $pCO_2$. This, in turn, reduces the required $CO_2$ injections to reach and maintain the 1.5°C climate target. Dissolution of $CaCO_3$ deep-sea sediments caused by the injection of $CO_2$ into the deep ocean at
3000m causes the dissolution of 11.8 Pmol $CaCO_3$ in the $A2_{sed}$ simulation by year 3020 releasing 11.8 Pmol DIC and 23.6 Pmol alkalinity to the deep ocean. Highest $CaCO_3$ dissolution rates occur in the vicinity of the seven injection sites (Figs. S4 a, b). Hence, ocean acidification reaching the deep ocean and ocean CCS convert sediments from a $CaCO_3$ sink (116 Gt Ca-$CO_3$-C at the end of the respective spin-up run) to a source of its dissolution products. A second process that contributes to the increase in ocean alkalinity is the terrestrial $CaCO_3$ weathering flux which arrives in the surface ocean via river dis-
charge, and amounts to about 19.3 Pmol alkalinity and 9.7 Pmol C (116 Gt C) by the end of the $A2_{sed}$ simulation.

Disentangling the relative role of the two processes (turning $CaCO_3$ burial into $CaCO_3$ dissolution; continuous flux of alkalinity from terrestrial weathering) with respect to stabilizing oceanic $CO_2$ uptake and thereby affecting the required $CO_2$ injections is not trivial. Waters affected by $CaCO_3$ sediment dissolution in the deep ocean need to return to the ocean surface before having an effect on surface ocean $pCO_2$ and oceanic $CO_2$ uptake (Cao et al., 2009). The fluxes from terrestrial
weathering, however, are in our simulation, continuous and constant with time (no sensitivity of weathering to changes in atmospheric $pCO_2$, surface air temperature, precipitation, or terrestrial production), and directly arrive in the surface ocean via river inflow. It is thus likely that, in comparison to the standard experiments without terrestrial weathering, the latter affects the atmosphere-ocean $CO_2$-flux well before the alkalinity input related to $CaCO_3$ dissolution. Quantifying the effect of each process to reduce the required $CO_2$ injection individually, however, would require additional simulations, e.g. experi-
ments with $CaCO_3$ dissolution turned on but terrestrial weathering turned off. This is beyond the scope of this study. In con-



sequence, of the two processes mentioned above, the required emission reduction amounts to about 846 Gt C, i.e. ~ 109 Gt C

(11 %) less when compared to the *A2* run (Fig. 2 b).

The net effects of sediment/weathering feedbacks on the required $CO_2$ injections in simulations of the second approach described above are as well represented in the injection experiments of the first and third approach, but are of smaller

magnitude, i.e., 5 % less (Fig. 2 a, b, c).

### 3.3 Biogeochemical impacts

Here, we present injection related biogeochemical impacts with respect to changes in pH and the saturation state of

aragonite in the default simulations of the second (*A2*) and third approach (*A3*) and of the respective RCP 4.5 control run.

Simulations of the first approach are neglected here, because this one cannot reach and maintain the 1.5°C climate target.

At the beginning of our default simulations (year 2020), the uptake of anthropogenic $CO_2$ has lowered average pH

at the ocean surface by about 0.12 units, relative to its preindustrial value of about 8.16 (Fig. 3 a). This trend continues in the

control simulation until its maximum reduction of about -0.25 units in the year 2762, which stays nearly constant until the

end of the simulation (Fig. 3 a).

As direct $CO_2$ injections lead to a decline in the atmospheric $CO_2$ concentration (Fig. 1 c) and, in consequence, to a

lower upper-ocean carbon uptake via air-sea gas exchange, we find smaller reductions in average ocean surface pH, i.e. reduced upper ocean acidification, after year 2045 in the *A2* simulation and after year 2031 in the *A3* run (Fig. 3 a), i.e., shortly

after their respective starting points of oceanic CCS (Fig. 1 a). In year 3020 the average ocean surface pH in the *A2* simulation is about +0.13 units higher, when compared to the control run (Fig. 3 a). Using global mean surface ocean pH as a metric, surface ocean acidification in year 3020 compared to year 2020 is slightly more intense in the *A2* simulation, but even

more reduced in the *A3* run.  In both cases this is a direct effect of a lower atmospheric $pCO_2$ (Fig. 1 c) compared to year

2020. Amelioration of surface ocean pH shows regional variability (Fig. 3 b), with local maxima of the pH difference between the *A2* simulation and the control run in the year 3020 up to +0.23 units, in particular in northern latitudes (Fig. 3 b).

However, surface ocean acidification is less reduced in the Southern Ocean and even slightly higher in parts of the Weddell

Sea, where most of the injected $CO_2$ leaks back into the atmosphere (Fig. 3 b).

The simulated ameliorations in the surface ocean pH come at the expense of strongly acidified water masses in the

vicinity of the seven injection sites at 3000m depth, when compared to the RCP 4.5 control run. In order to assess how much

of the global ocean volume ($\sim 1.6 \times 10^9$ $km^3$) shifts to biotically critical pH values in our simulations, we define two pH categories. The first category is defined as $7.4 \leq pH \leq 7.8$ (solid lines in Fig. 3 c) and is chosen because studies have shown that

all calcifiers such as coralline algae and foraminaferans are strongly reduced or are absent from acidified areas (pH < 7.8)

and the overall biomass of the benthic community is about 30 % less compared to normal conditions (e.g., IPCC, 2011; Fabricius et al., 2015). The second category includes pH values that are lower than 7.4 (dashed lines in Fig. 3 c). Such low pH



values are for instance found in the vicinity of volcanic $CO_2$ vents and cause a massive drop in biodiversity (e.g., Ogden, 2013).

In our control simulation, we find a steady increase in the ocean volume characterized by $7.4 \leq pH \leq 7.8$, from
about 11 % of total ocean volume in year 2020 to about 63 % in year 3020 (Fig. 3 c). Oceanic CCS in the *A2* and *A3* simula-

tion leads to a much steeper increase of 'moderately' acidified waters ($7.4 \leq pH \leq 7.8$) with maximum values of 76 % and 71

%, respectively, in year 2551 (Fig. 3 c), but decreasing to 72 % and 64 % in year 3020. Considering our chosen category ($7.4$

$\leq pH \leq 7.8$) ocean CCS mainly speeds up interior-ocean acidification but does not increase the acidified volume at the end of

the simulation very much. At the end of the simulation the *A3* simulation and the *A2* run show an increase of affected interi-
or-ocean water by 1 and 13 %, respectively, compared to the control run.

Respective volumes of the second category ($pH < 7.4$) start to appear around the year 2400 in the control simulation and then

slowly increase to about 2 % until the end of the simulation (Fig. 3 e). In contrast, oceanic CCS directly results in the imme-

diate appearance of waters with $pH < 7.4$, with a volume steadily increasing until the year 3020 where it reaches 9 % of total

ocean volume in the *A2* simulation and 15 % in the *A3* run (Fig 3 c). The differences in both pH categories between the in-
jection experiments are due to the higher cumulative mass of injected $CO_2$ in the *A3* run, leading to a smaller volume in the

first category and to a bigger volume in the second one (Figs. 1 a, 2 b).

In order to further identify extreme pH related to the injections, we look at minimum pH values. These are found at 3000m

depth, i.e., the depth at which oceanic CCS is carried out. Relative to preindustrial conditions, the highest reductions in pH

minima are found in the *A2* simulation with about -2.37 units in year 2062, however with large regional variability (Fig. 3 d,
e). Subsequently, the pH minima in the *A2* simulation show strong oscillations until about the year 2400, which are caused

by the different annual $CO_2$ injection rates. By the end of the *A2* simulation, minimum pH values at 3000m depth are up to 1

unit lower than in the control run (Fig. 3 d). We find a similar pattern in the *A3* simulation, although the pH reductions show

only slight oscillations, resulting in a more constant pH reduction than in the *A2* simulation (Fig. 3 d). In comparison to the

injection experiments, minimum pH values in the control run start to appear from the year 2300 onwards, leading to a reduc-
tion by about -0.17 units in the year 3020 (Fig. 3 d), i.e. the deep ocean feels ocean acidification very slowly.

To summarize we observe an increasing benefit in reduced acidification at the ocean surface with higher cumulative

$CO_2$ injections, which comes at the expense of increasing acidified water masses in the intermediate and deep ocean with

strongest pH reductions in the vicinity of the injection sites (Fig. 3 e). Figure 4 a, b illustrates this trade-off for the injection

experiments of the *A2* and *A3* simulations as well as for the respective control runs in year 3020. By comparing the different
simulations with each other, we find that continental weathering and $CaCO_3$ sediment feedbacks lead to a slightly higher

increase in average pH at the ocean surface as well as smaller minimum pH values at 3000m depth, when compared to pre-

industrial. This is caused by the dissolution of $CaCO_3$ sediments and the terrestrial weathering flux, which both have the net

effect of adding alkalinity to the ocean and thereby increasing the buffer capacity of seawater.

The reported reductions in global average surface pH in our control simulation caused by the partial oceanic uptake

of the RCP 4.5 $CO_2$ emissions correspond to an increase in hydrogen ions ($H^+$), which partly react with carbonate ions

($CO_3^{2-}$) to form bicarbonate ions ($HCO_3^-$). This leads in consequence to a reduction in the surface saturation state ($\Omega$) with

respect to the $CaCO_3$ minerals aragonite and calcite. This is of importance to marine calcifiers, because the formation of

shells and skeletons generally occurs where $\Omega > 1$ and dissolution occurs where $\Omega < 1$ (unless the shells or skeletons are pro-

tected, for instance, by organic coatings) (Doney et al., 2009; Guinotte and Fabry, 2008). Since aragonite is about 1.5 times

more soluble than calcite (Mucci, 1983) and since aragonite is the mineral form of coral reefs, which are of large socio-

economic value, we only report here on simulated changes in the saturation state of aragonite.

To investigate how tropical coral reef habitats might be impacted in our simulations, we here define the potential

coral reef habitat as the volume of the global upper ocean (0:130 m, the two topmost model grid cells), which is character-

ized by $\Omega_{AR} > 3.4$ and ocean temperatures between 21°C and 28°C, where most coral reefs exist (Kleypas et al., 1999). We

present this volume as the percent fraction of the total upper ocean volume ($4.637 \times 10^7$ km$^3$) in our model.

For preindustrial conditions (year 1765), we find that about 37 % of the upper ocean volume is within our defined

thresholds (green star in Figs. 5 a, b). At the beginning of our simulations (year 2020), this coral reef habitat volume has al-

ready declined to about 13 %, consistent with the current observation that many coral reefs are already under severe stress

(e.g., Pandolfi et al., 2011; Ricke et al., 2013). In the RCP 4.5 control run, we observe that the potential tropical coral reef

habitat volume reaches 0 % in the year 2056 and remains so thereafter (Fig. 5 a) with a decrease in aragonite oversaturation

levels being the main driver.

In our injection experiments, we find an increase in the potential tropical coral reef habitat volume right after the

start of oceanic CCS (Fig. 5 a).  Despite this, in the *A2* simulation the respective volume still approaches zero (0.2 %) in the

year 2044, although it does then steadily increase again until it reaches 21 % at the end of the simulation, i.e. still 16 % less

than its preindustrial state, but also 8% more compared to the current situation (Figs. 5 a, b). The respective habitat volume

in the *A3* simulation shows an earlier and stronger increase, resulting in a habitat volume of about 34 %, i.e. 3 % less than

preindustrial, at the end of the model experiment (Fig. 5 a).

In preindustrial times, water masses in the upper ocean (0 - 130 m) that were undersaturated with respect to arago-

nite ($\Omega_{AR} < 1$) were negligible (0.2 %; Fig. 5 c, green star). This undersaturated volume has increased to about 1 % at the

beginning of our simulations. Over the course of the control run, we observe an increase with a maximum of about 9 % in

the year 2212. Subsequently, the respective undersaturated volume slightly decreases until it reaches a minimum of about 7

% at the end of the simulation. Undersaturated surface waters are located in higher latitudes (Fig. 5 d), which is for instance

considered a threat to pteropods like Limacina helicina (e.g., Lischka et al., 2011). In the *A2* and *A3* simulations, the respec-

tive undersaturated volumes are significantly smaller and never exceed 2 % of the surface ocean volume (Fig 5 c, d). Under-

saturated surface-water volumes in the *A2* run are slightly higher than those in the *A3* simulation.



Further, we assess the volume that is undersaturated with respect to aragonite in the intermediate and deep ocean

(130 - 6080 m) and present it as a %-fraction of the entire interior ocean volume ($1.311 \times 10^9$ km$^3$). This is of interest since

changes in interior-ocean $\Omega_{AR}$ may affect the growth conditions of cold-water corals (e.g., Guinotte et al., 2008; Flögel et al.,

2014; Roberts and Cairns, 2014) and the dissolution depth of sinking aragonite particles.

At the beginning of the simulations, 69 % of the interior oceans are undersaturated with respect to aragonite, which

is about 3 % more than preindustrial (Fig. 5 e). Subsequently, the increase in undersaturated water volume is similar among

all simulations until about the year 2122, when the undersaturated volume in the control simulation continues to increase

until its maximum of about 91 % in the year 2713. The undersaturated volumes in the injection experiments show only a

very small increase after year 2122, leading until year 3020 to values of about 86 % in both injection simulations (Fig. 5 e).

The bigger volume in the control run is likely caused by acidified waters at the ocean surface that ventilate intermediate and

mode waters (Resplandy et al., 2013).

Figs. 6 a, b shows a similar trade-off in the injection experiments of the second and third approach in year 3020 as

for pH (Figs. 4 a, b), i.e. an increase of the aragonite saturation states in the upper ocean and an increase of undersaturated

conditions in the intermediate and deep ocean. Further, the effects of $CaCO_3$ sediment dissolution and continental weather-

ing lead to the highest benefit in the upper ocean and the lowest harm in the intermediate and deep ocean (Figs. 6 a, b).

As mentioned in the introduction, the neglect of non-$CO_2$ greenhouse gases in our injection experiments underesti-

mates the required cumulative $CO_2$ injections and associated trade-offs in each approach. This is due to the fact that non-$CO_2$

greenhouse gases directly affect the Earth's energy balance, resulting in either warming or cooling of the atmosphere. Gases

like methane and nitrous oxide warm the Earth, while aerosols such as sulfate cool it (e.g., Myhre et al., 2013). The current

net effect is a small positive radiative forcing, which, although controversially debated, is expected to increase as the cooling

effect of sulfate aerosols is predicted to decline over the next half of this century (Moss et al., 2010; Hansen et al., 2017; Rao

et al., 2017).

**4. Conclusion**

This modeling study explores the potential and biogeochemical impacts of three different approaches to control the

amount of oceanic CCS as a means to fill the gap between the $CO_2$ emissions and climate impacts of the RCP 4.5 scenario

and a specific temperature target such as the 1.5°C climate target. We do so from the perspective of using only ocean CCS

for this purpose.

The analysis of the *A1* simulation (first approach) reveals that because of committed warming and eventually out-

gassing of some of the injected $CO_2$, it would not be sufficient to inject the residual of the RCP 4.5 $CO_2$ emissions (964 Gt C

in total) until the year 3020 once a global mean temperature of 1.5°C is exceeded for the first time (year 2045). In order to

overcome the observed overshoot of +0.7°C by year 3020 in the first approach, we find that about 600 Gt C (62 %) more



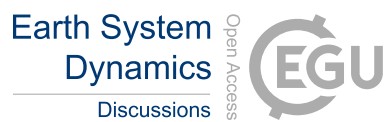

need to be injected, as indicated by the default simulation of the second approach, i.e., *A2* run (Figs. 1 a, b, 2 b).

To follow the atmospheric $CO_2$ concentration of the RCP/ECP 2.6 as closely as possible by applying oceanic CCS would require cumulative $CO_2$ injections of about 2200 Gt C until the year 3020. However, global mean temperature reaches
+0.9°C by the end of the *A3* simulation and thus undershoots the respective climate target.

The cumulative $CO_2$ injections in the second and third approach and the respective required emission reductions questions the suitability of oceanic CCS for the aspired target on such a timescale, because the outgassed $CO_2$ amounts, which are 607 and 900 Gt C by year 3020, respectively (Figs. 1 d, 2 b, c), would need to be re-captured by additional technologies such as Direct Air Capture and subsequently re-injected into the deep ocean. The required emission reductions of
about 955 Gt C in the second and about 1300 Gt C in the third approach, point to the massive $CO_2$ amounts that would need to get removed from the atmosphere under the RCP/ECP 4.5 $CO_2$ emission scenario in order to be compatible with the 1.5°C or lower climate target on a millennium timescale.

From the integrated analysis of the model runs from all three approaches (i.e. eye-fitted lines in Figs. 2 b, c), we quantify the amount of emission reduction and oceanic CCS, respectively, required to lower the model-predicted global
mean temperature by 1°C. In the near-term (2100) this measure is 446 Gt C / 1°C for both oceanic CCS and the required emission reductions as only a tiny fraction has outgassed until that point in time (see section 3.1). On a millennial timescale this measure is about 951 Gt C / 1°C for oceanic CCS and about 595 Gt C / 1°C (37 % less) for the required emission reductions, respectively, highlighting that a large fraction of injected $CO_2$ has outgassed.

Inclusion of $CaCO_3$ sediment and weathering feedbacks reduces the required cumulative $CO_2$ injections and re-
quired emission reductions by about 6 % in the first and third approach and by about 11 % in the second approach, respectively (Fig. 2 b, c). The neglect of non-$CO_2$ greenhouse gases in the applied forcing of the injection experiments underestimates the cumulative $CO_2$ injections that would be required. In general, it is estimated that non-$CO_2$ climate agents contribute between 10-30 % of the total forcing (Friedlingstein et al., 2014) until the year 2100 and for business-as-usual simulations. Extrapolating the current contribution of greenhouse gases other than $CO_2$ qualitatively into the future we expect that
$CO_2$-injections of the magnitude of the *A3* simulation may be required to stay safely below +1.5°C on a millennium timescale. We propose, that our generalized estimates of emission reduction and oceanic CCS per 1°C cooling, respectively, may be used in the future to quantify additional efforts in order to compensate for non-$CO_2$ greenhouse gases induced warming.

With respect to the biogeochemical impacts in the injection simulations of the second and third approach, we observe an increase of average pH and aragonite saturation states in the surface ocean (0 - 130 m) after the start of oceanic
CCS, when compared to the RCP 4.5 control run. These are due to the direct effect of a lower atmospheric $pCO_2$ in the injection experiments, i.e., reduced upper ocean acidification (section 3.3).

Potential tropical coral reef habitats in the upper ocean volume, which are here defined as $\Omega_{AR} > 3.4$ and ocean temperatures between 21°C and 28°C, are observed to steadily increase after the start of oceanic CCS in the *A2* run and the

*A3* simulation (Fig. 5 a), almost reaching preindustrial levels in the *A3* simulation. However, the potential coral reef habitats

in the respective injection experiments are close to zero for several decades (Fig. 5 a), raising the question if coral reefs

would be able to recover from globally inhabitable conditions after this period of time. Local application of ocean alkaliniza-

tion (Feng et al., 2016) may be a technical solution to protect coral reefs during this time period, in particular in regions

where coral reefs are essential for shoreline protections.

The observed reduction of ocean acidification in the surface ocean comes at the expense of more strongly acidified

water masses in the intermediate and deep ocean, with strongest reductions in pH in the vicinity of the seven injections sites

(Figs. 3 d, e). Although it is difficult to predict how this would impact marine ecosystems, it is very likely that such condi-

tions would put them under severe stress.

Overall, the trade-off between injection-related damages in the deep ocean and benefits in the upper ocean illustrate

the challenge of evaluating the offset of local harm against global benefit, which is very likely the subject of any deliberate

$CO_2$ removal method (e.g., Smith et al., 2016; Boysen et al., 2017; Fuss et al., 2018). Leaving aside the massive economic

effort associated with ocean CCS of the size needed to reach the 1.5°C climate target (even when starting from a currently

optimistic RCP 4.5 mitigation scenario), humanity will have to decide whether severe stress and potential loss of deep-sea

ecosystems is acceptable when paid off by conserving or restoring surface ocean ecosystems to a large extent.






**Data availability**

The model data used to generate the table and figures is available online at

https://data.geomar.de/thredds/catalog/open_access/reith_et_al_2019_esd/catalog.html

**Author contributions**

All authors conceived and designed the experiments. FR implemented the experiments with contributions from WK and JG. FR performed the experiments and analysed the data. FR wrote the manuscript with contributions from all co-authors.

**Competing interests**

The authors declare that they have no conflict of interest.

**Acknowledgments**

The Deutsche Forschungsgemeinschaft (DFG) financially supported this study via the Priority Program 1689.







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





**Table 1:** Overview of all conducted simulations and their set-up. The "X" denotes that the respective feature is applied. Note that the applied $CO_2$ forcing follows the RCP 4.5 $CO_2$ emission scenario from 2006-2100 and the Extended RCP 4.5 $CO_2$ emissions scenario from 2100-2500. From 2500 onwards $CO_2$ emissions linearly decline until zero Gt C yr$^{-1}$ in year 3020.

| Simulation | Set-up | | |
|---|---|---|---|
| | $CO_2$ emissions forcing | $CaCO_3$ sediment and weathering feedbacks | Direct $CO_2$ injections at 3000 m depth |
| | 2006-3020 | 2006-3020 | 2020-3020 |
| RCP 4.5 control run | X | | |
| A1 | X | | X |
| A1_Comitw | X | | |
| A2 | X | | X |
| A3 | X | | X |
| RCP 4.5 control$_{sed}$ run | X | X | |
| A1$_{sed}$ | X | X | X |
| A1_Comitw$_{sed}$ | X | X | |
| A2$_{sed}$ | X | X | X |
| A3$_{sed}$ | X | X | X |









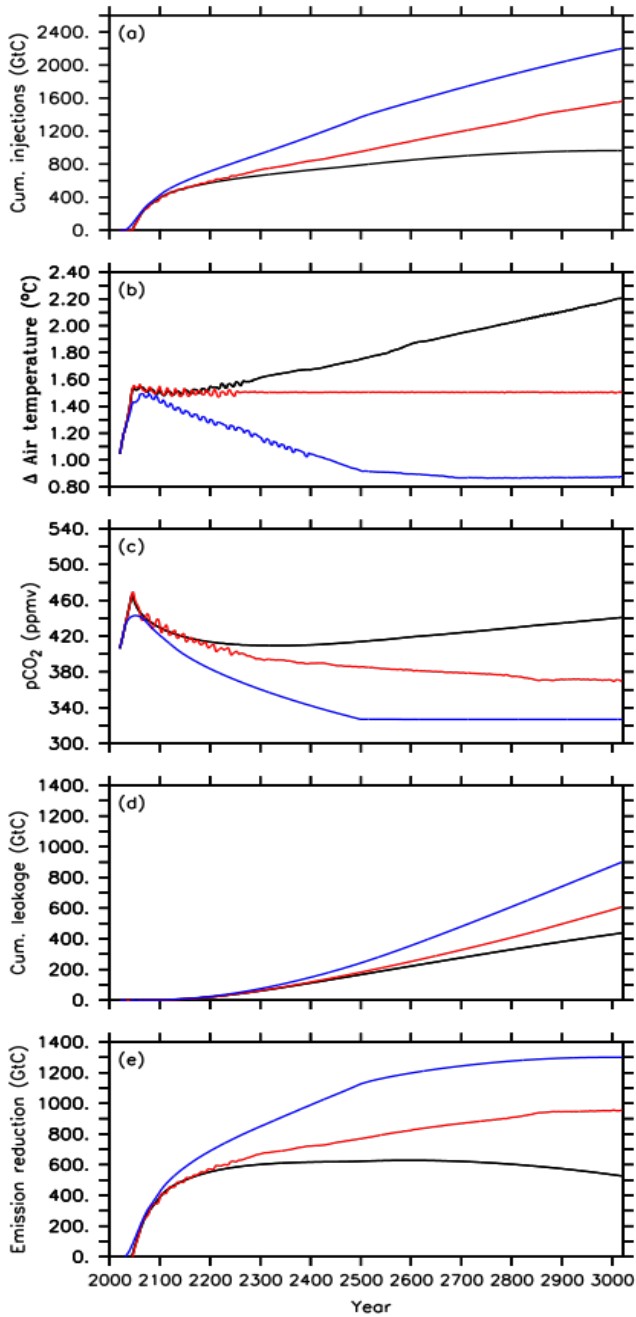

**Figure 1:** Time-series of the different default injection experiments, i.e., A1 simulation (black lines), A2 simulation (red lines) and A3 simulation (blue lines) for **(a)** cumulative $CO_2$ injections, **(b)** global mean surface air temperature, relative to preindustrial, **(c)** atmospheric $CO_2$ concentration, **(d)** cumulative leakage of injected $CO_2$, and **(e)** required emission reduction.

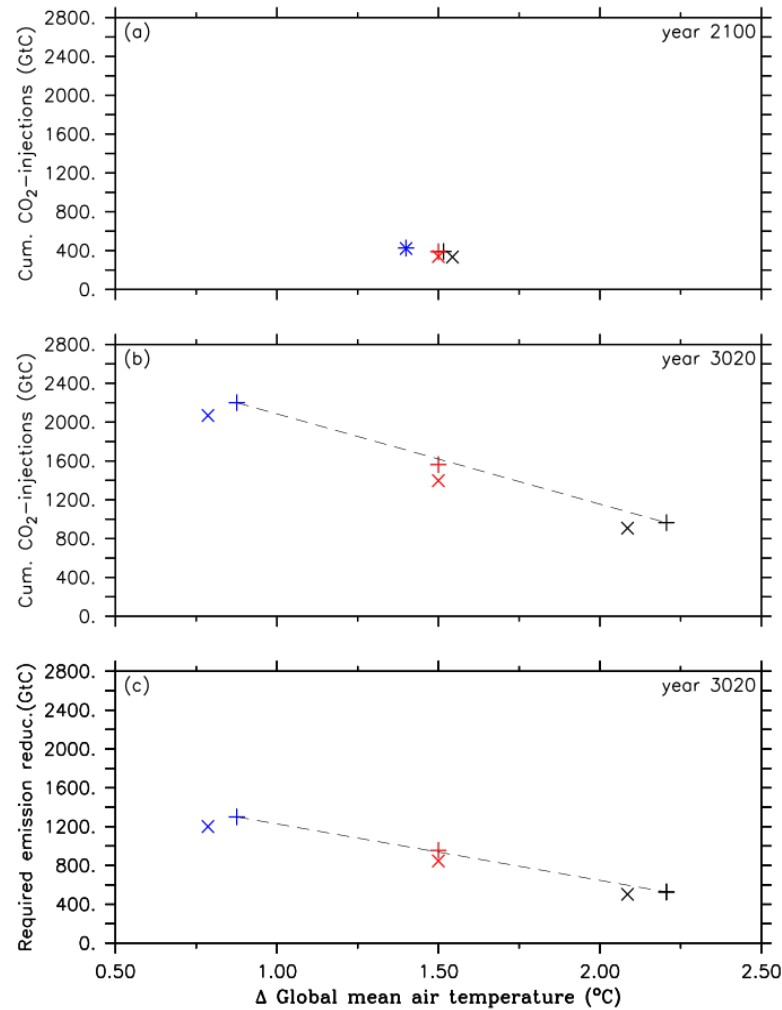

**Figure 2:** Comparison between simulations of the first approach (A1, black symbols), simulations of the second approach (A2, red symbols) and simulations of the third approach (A3, blue symbols). The cross symbols refer to the default simulations and the X symbols denote simulations with $CaCO_3$ sediment and weathering feedbacks. These symbols represent for **(a)** cumulative $CO_2$ injections and corresponding global mean temperature, relative to preindustrial, in year 2100 **(b)** cumulative $CO_2$ injections and corresponding global mean temperature, relative to preindustrial, at the end of the simulation (yr 3020), and **(c)** required emission reduction and corresponding global mean temperature, relative to preindustrial, at the end of the simulations. Note that the dashed black lines are eye-fitted to the results of the standard runs.




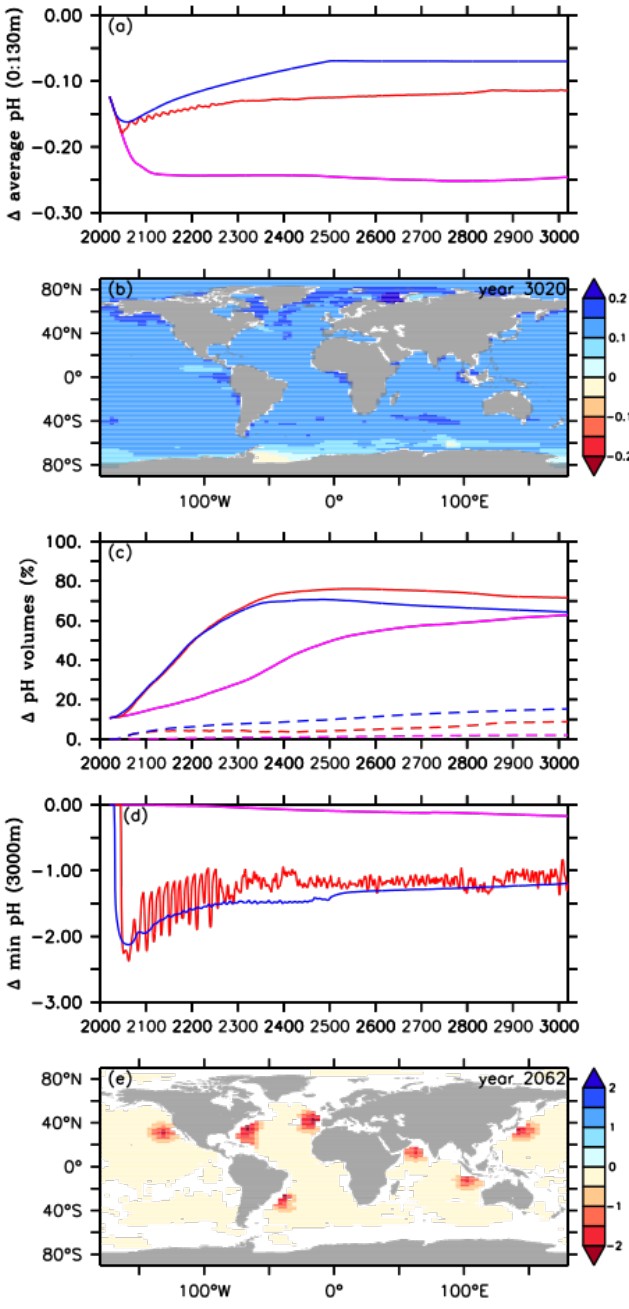

**Figure 3:** Comparison of pH values between the default RCP 4.5 control run (purple lines), the A2 simulation (red lines) and the A3 simulation (blue lines) for **(a)** average ocean surface pH (0 to 130 m depth), **(b)** difference in ocean surface pH, relative to preindustrial, between the A2 simulation and the default RCP 4.5 control run in yr 3020, **(c)** pH volumes of first (≤ 7.8 and ≥ 7.4, solid lines) and second category (< 7.4, dashed lines), **(d)** minimum pH values at 3000 m depth, and **(e)** difference in minimum pH at 3000 m depth between the A2 simulation and the default RCP 4.5 control run in yr 2062.


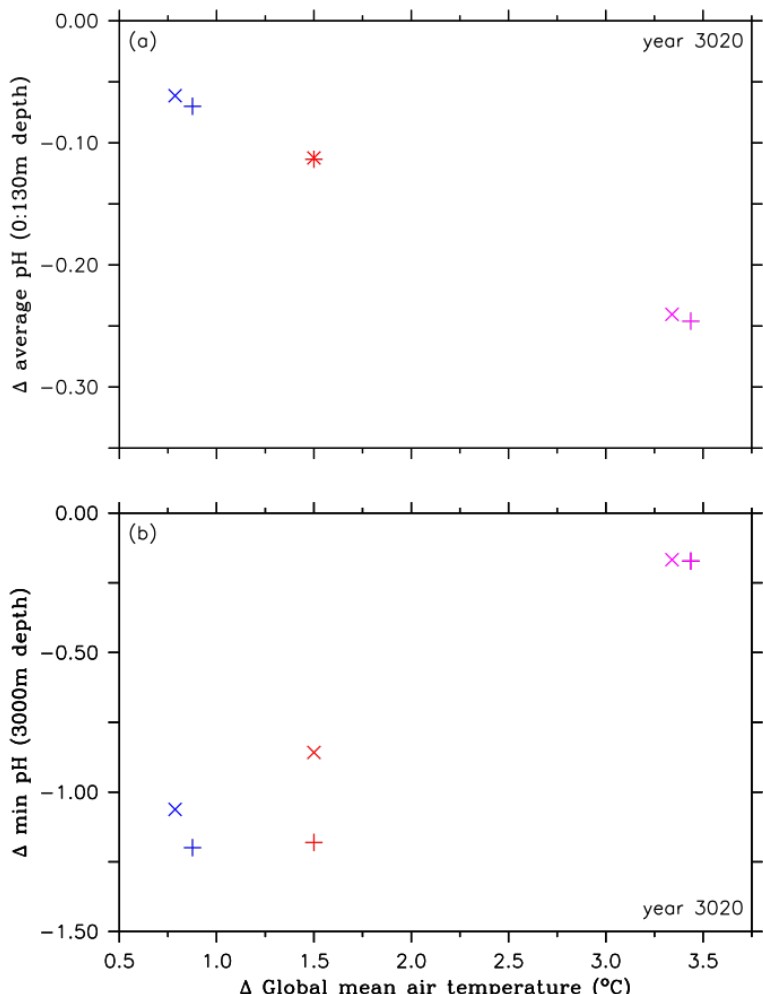

**Figure 4:** Comparison of pH values and corresponding global mean temperature in the year 3020, both relative to preindustrial, between the RCP 4.5 control simulations (purple symbols), simulations of the second approach (A2, red symbols) and simulations of the third approach (A3, blue symbols). The cross symbols refer to the default simulations and the X symbols denote simulations with $CaCO_3$ sediment and weathering feedbacks. These symbols represent for **(a)** changes in ocean surface pH (0 to 130m depth), relative to preindustrial, and **(b)** changes in minimum pH values at 3000 m depth, relative to preindustrial.





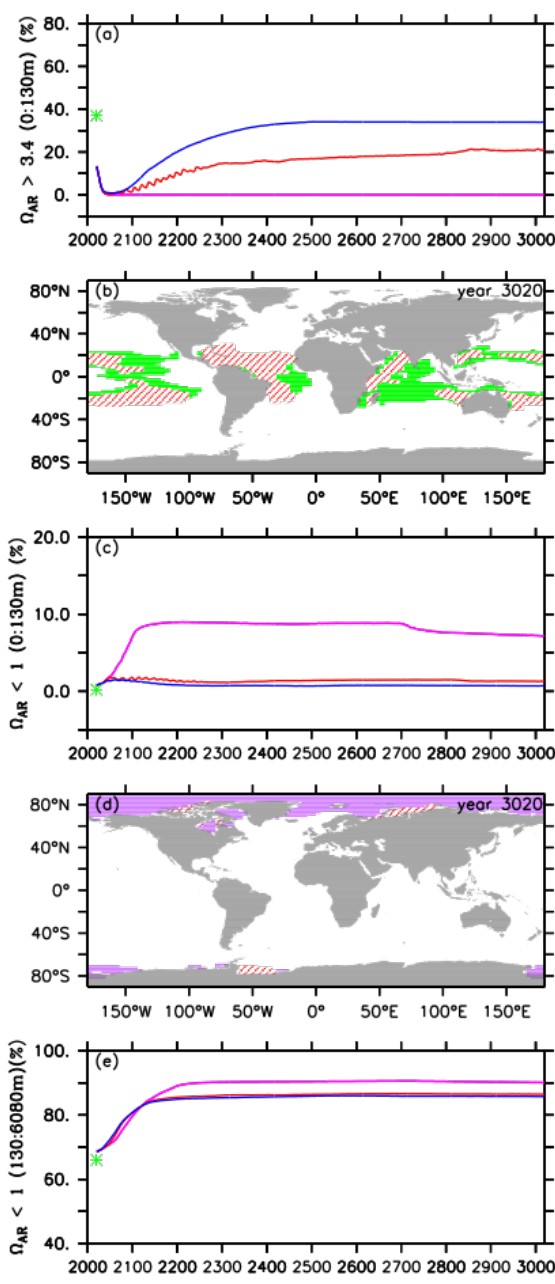

**Figure 5:** Comparison of volumes for different saturation states between preindustrial (green stars), the default RCP 4.5 control run (purple lines), the A2 simulation (red lines), and the A3 simulation (blue lines) for **(a)** omega aragonite > 3.4 in the upper ocean (0 to 130m depth), **(b)** potential coral reef habitat defined as the volume of the global upper ocean (0 to 130m depth) with omega aragonite > 3.4 and ocean temperatures between 21°C and 28°C for preindustrial (green) and the A2 simulation (red hatching) in in the year 3020, **(c)** omega aragonite < 1 in the upper ocean (0 to 130m depth), **(d)** global distribution of omega aragonite < 1 for the default control run (purple) and the A2 simulation (red hatching), and **(e)** omega aragonite < 1 in the intermediate and deep ocean (130 to 6080m depth).


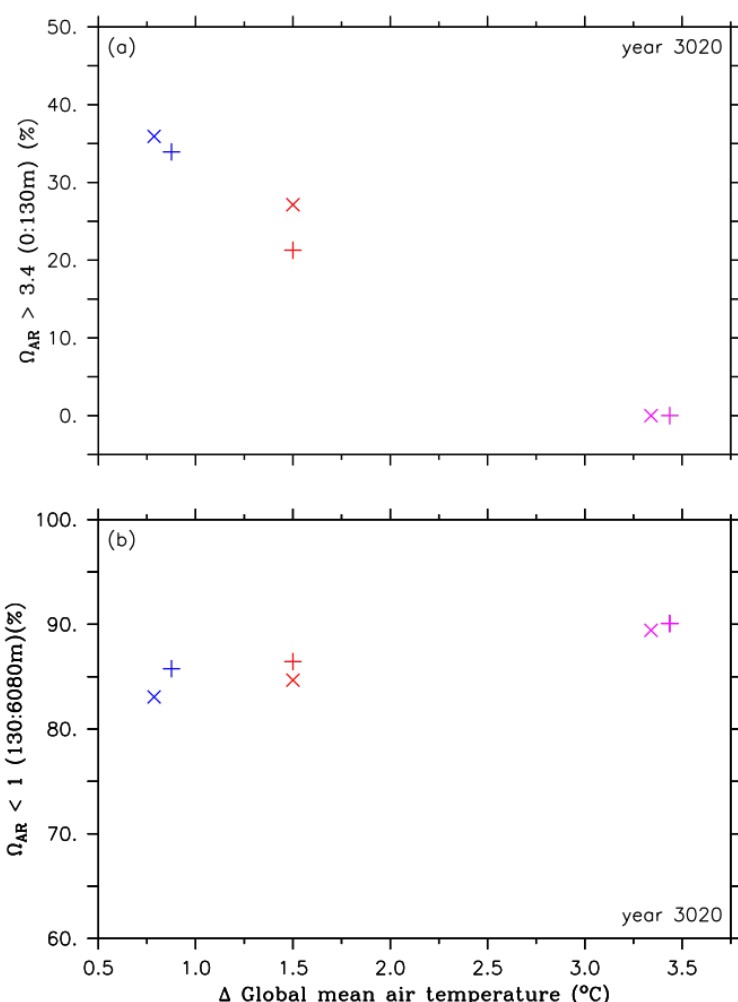

**Figure 6:** Comparison of volumes for different aragonite saturation states and corresponding global mean temperature in the year 3020, both relative to preindustrial, between the RCP 4.5 control simulations (purple symbols), simulations of the second approach (A2, red symbols) and simulations of the third approach (A3, blue symbols). The cross symbols refer to the default simulations and the X symbols denote simulations with $CaCO_3$ sediment and weathering feedbacks. These symbols represent for **(a)** omega aragonite > 3.4 in the upper ocean (0 to 130m depth), relative to preindustrial, and **(b)** changes in minimum pH values at 3000 m depth, relative to preindustrial.