# Peer review of "Figure S1: Global mean surface air temperature, relative to preindustrial, of the A1\_Comitw simulation (solid) and the A1\_Comitw\_sed run (dashed). The horizontal dashed black line denotes the 1.5°C climate target."

_Earth System Dynamics, 2018_

## Referee Comment (RC1) · Anonymous Referee #1 · 3 Apr 2019

In this study, the authors used an Earth system model of intermediate complexity, UVic, to investigate the effect of direct CO2 injection on atmospheric CO2, temperature, and ocean acidification. Under the reference CO2 emission scenario of RCP4.5, CO2 is injected directly in the form of dissolved inorganic carbon into deep sea sites around 3000m to limit global warming below 1.5 K. Three injection strategies are designed and simulated: 1) once global mean warming exceeds 1.5K, all further emission are injected into the deep ocean; 2) inject CO2 into the deep ocean in an amount to keep global mean warming below 1.5K; 3) inject CO2 into the deep ocean in an amount such that atmospheric CO2 follows the concentration pathway of RCP 2.6. In all injection scenarios, substantial amount of CO2 outgassing is found, indicating that a significant amount of carbon leakage needs to be re-injected to achieve the mitigation goals. With

respect to the biogeochemical effects, it is found that CO2 injection into the deep ocean mitigate surface acidification, but at the cost of more acidified water in the deep ocean, in particular in the vicinity of the injection sites. This study is well-defined and the analysis is comprehensive and in-depth. This study makes a useful contribution to the assessment of climate and environmental effect of direct CO2 injection.

I have a few suggestion that the authors may want to take into consideration:

1) The mitigation scenario of RCP 4.5 is used as the reference scenario for all injection experiments. It would be illustrative to demonstrate the effect of direct CO2 injection under higher emission scenarios (i.e., RCP 8.5). To achieve the same mitigation goals, deep ocean would be much more acidified, and presumably, more CO2 would be out-gassed from the deep ocean.

2) In terms of deep ocean acidification, the authors may want to look at the evolution of aragonite (calcite) saturation horizon.

3) What is the role of land carbon cycle here? How much of the outgassed CO2 can be attributed to (or compensated by) the terrestrial CO2 flux? Can the land carbon cycle feedback be quantified by turning on/off the UVic land carbon component?

———————————————————

---

## Referee Comment (RC2) · Anonymous Referee #2 · 4 Apr 2019

CO2 emissions are increasing at an unprecedented rate into the earth's atmosphere. By and large, global political leadership have recognized the consequences of such emissions for human kind and ecosystems. As a result, the 2015 Paris Agreement has set the target of limiting global warming to below 2°C. To achieve such a target, academicians have been discussing some unconventional methods – known as geo-engineering. To the same effect, in this study, Reith and co-workers have presented this excellent and very thorough analysis of consequences of injecting atmospheric CO2 into the deep oceans. Their analysis looks robust (I am not a modeller though!). I have just a couple of comments that might be discussed in the revised version of the manuscript:

1. I am not sure if the trade-off between the amount of CO2 released back to the at-

mosphere in collecting CO2 from the atmosphere and injecting it into the deep CO2 has been considered. By which way(s) the atmospheric CO2 can be collected from the atmosphere and put into the deep ocean, and how much CO2 will be emitted back (through the instruments used for such huge task) to achieve both the actions. I know this might not be feasible to incorporate in the model but it needs to be discussed/mentioned.

2. Can (gas chromatographically) CO2 alone be collected from the atmosphere on such a large scale? Or will CO2 be part of the mixture of all the atmospheric gases and particles (aerosols)? Was the model tuned for injecting of natural air rather than only CO2 into the deep ocean? How sensitive mixture would become for ocean chemistry?

---

## Author Comment (AC2) · 28 Jul 2019

*First of all, the authors thank the reviewer very much for her/his thoughtful and constructive suggestions.*

In this study, the authors used an Earth system model of intermediate complexity, UVic, to investigate the effect of direct $CO_2$ injection on atmospheric $CO_2$, temperature, and ocean acidification. Under the reference $CO_2$ emission scenario of RCP4.5, $CO_2$ is injected directly in the form of dissolved inorganic carbon into deep sea sites around 3000m to limit global warming below 1.5 K. Three injection strategies are designed and simulated: 1) once global mean warming exceeds 1.5K, all further emission are injected into the deep ocean; 2) inject $CO_2$ into the deep ocean in an amount to keep global mean warming below 1.5K; 3) inject $CO_2$ into the deep ocean in an amount such that atmospheric $CO_2$ follows the concentration pathway of RCP 2.6. In all injection scenarios, substantial amount of $CO_2$ outgassing is found, indicating that a significant amount of carbon leakage needs to be re-injected to achieve the mitigation goals. With respect to the biogeochemical effects, it is found that $CO_2$ injection into the deep ocean mitigate surface acidification, but at the cost of more acidified water in the deep ocean, in particular in the vicinity of the injection sites. This study is well-defined and the analysis is comprehensive and in-depth. This study makes a useful contribution to the assessment of climate and environmental effect of direct $CO_2$ injection. I have a few suggestions that the authors may want to take into consideration:

1) The mitigation scenario of RCP 4.5 is used as the reference scenario for all injection experiments. It would be illustrative to demonstrate the effect of direct $CO_2$ injection under higher emission scenarios (i.e., RCP 8.5). To achieve the same mitigation goals, deep ocean would be much more acidified, and presumably, more $CO_2$ would be outgassed from the deep ocean.

*Thank you for this very interesting point. We have also considered conducting a direct $CO_2$ injection (oceanic CCS) experiment under the RCP 8.5 $CO_2$ emission scenario such as in the study of Reith et al. (2016) in order to compare the Earth system response to the ones under the RCP 4.5 $CO_2$ emission scenario. However, we feel that such an oceanic CCS experiment does not really fit into the framing of our study, i.e., meeting climate targets. Any future emission trajectory close to the RCP 8.5 $CO_2$ emission scenario assumes that almost no climate policies are in place, implying that relatively costly oceanic CCS would not be undertaken. Oceanic CCS with estimated cost units of about 100 USD/t$CO_2$ (IPCC, 2005) would only be considered as an option if other less costly emission mitigation options such as switching to renewable energies and increasing insulation in houses have been fully utilized so that $CO_2$ emissions have been reduced to coincide with the RCP 4.5 or even RCP 2.6 $CO_2$ emission scenario. Only $CO_2$ emission mitigation scenarios like the RCP 4.5 or in particular the RCP 2.6 $CO_2$ emission scenario imply implicit carbon prices above 100 USD/t$CO_2$ (Clarke et al., 2014), making options like oceanic CCS attractive to supplement conventional $CO_2$ emission control such as the National Determined Contributions of the Paris Agreement. Accordingly, we would like to restrict our analysis to the implementation of oceanic CCS to climate policy relevant scenarios.*

2) In terms of deep ocean acidification, the authors may want to look at the evolution of aragonite (calcite) saturation horizon.

*Thank you for this suggestion. We have looked at the evolution of the aragonite saturation horizon (ASH) in the RCP 4.5 control run and our A2 experiment. In comparison to the beginning of the RCP 4.5 control run (Fig. R1), the strongest shallowing of the ASH in the RCP 4.5 control run by the year 2100 appears in some down- and upwelling regions (Fig. R2), which is due to deeper penetration and higher concentrations of anthropogenic $CO_2$ (e.g., Orr et al., 2005).*

[Figure]

**Figure R1:** Depth of aragonite saturation horizon (ASH) at the beginning of the RCP 4.5 control run (year 2020).

[Figure]

**Figure R2:** Relative changes in depth of aragonite saturation horizon (ASH) in the RCP 4.5 control run in 2100, i.e., year 2100 minus year 2020.

*With respect to the A2 experiment, we find an accelerated shoaling of the ASH in the Atlantic and smaller changes in the Pacific and Indian Ocean by the year 2100, when compared to the respective changes in the RCP 4.5 control run (Fig. R3). This can be explained by the fact that injections in the Pacific and Indian Ocean were carried out in waters already beneath the simulated ASH, which occurred at a depth of about 600 - 1200 m during the injection period (Fig. R1). This is in contrast to the Atlantic injection sites where the simulated ASH was at a depth of about 3000 m at the beginning of the simulated injections (Fig. R1), thus causing the simulated increase in the volume of undersaturated water. Consequently, the injections*

*could have, for example, additional implications for the distribution of aragonite forming deep-sea scleractinian corals (Guinotte et al., 2006) in the Atlantic. Furthermore, it is likely that the accelerated shoaling of the saturation horizons for aragonite (and calcite) in the Atlantic due to the injections would drive a change in habitat quality for a variety of deep-sea calcifiers (Orr et al., 2005).*

[Figure]

**Figure R3:** Relative changes in depth of aragonite saturation horizon (ASH) between the A2 simulation (year 2100) and the RCP 4.5 control run (year 2020).

*By the end of the simulations (year 3020), we observe similar changes in the ASH in the RCP4.5 control run when compared to the A2 experiment, although with a slightly higher shallowing of the ASH in the Pacific and Indian Ocean (Figs. R4-5).*

[Figure]

**Figure R4:** Relative changes in depth of aragonite saturation horizon (ASH) in the RCP 4.5 control run in 3020, i.e., year 3020 minus year 2020.

[Figure]

**Figure R5:** Relative changes in depth of aragonite saturation horizon (ASH) between the A2 simulation (year 3020) and the RCP 4.5 control run (year 2020).

*The authors feel that the biogeochemical part (section 3.3) in the submitted version of the manuscript is already very long and complex. To address this issue, we suggest adding the following paragraph and Figures R1-5 to the Supplement.*

*The new supplemental text reads:*

*"With respect to the evolution of the aragonite saturation horizon ($\Omega_{AR}$=1, ASH) after the beginning of the RCP 4.5 control run (Fig. S5), we observe that the ASH is mainly affected in down- and upwelling regions by the year 2100 (Fig. S6). This is caused by the deeper penetration and higher concentrations of the respective $CO_2$ emissions (e.g., Orr et al., 2005). With respect to the A2 experiment, we find an accelerated shoaling of the ASH in the Atlantic and smaller changes in the Pacific and Indian Ocean by the year 2100, when compared to the respective changes in the RCP 4.5 control run (Fig. S7). This can be explained by the fact that injections in the Pacific and Indian Ocean were carried out in waters already beneath the simulated ASH, which occurred at a depth of about 600 - 1200 m during the injection period (Fig. S5). This is in contrast to the Atlantic injection sites where the simulated ASH was at a depth of about 3000 m at the beginning of the simulated injections (Fig. S5), thus causing the simulated increase in the volume of undersaturated water. Consequently, the injections could have, for example, additional implications for the distribution of aragonite forming deep-sea scleractinian corals (Guinotte et al., 2006) in the Atlantic. Furthermore, it is likely that the accelerated shoaling of the saturation horizons for aragonite (and calcite) in the Atlantic due to the injections would drive a change in habitat quality for a variety of deep-sea calcifiers (Orr et al., 2005). By the end of the simulations (year 3020), we observe similar changes in the ASH in the RCP4.5 control run when compared to the A2 experiment, although with a slightly higher shallowing of the ASH in the Pacific and Indian Ocean (Figs. S8-9). We find nearly the same evolution pattern of the ASH in the A3 experiment (not shown)."*

*We have also added a sentence in the manuscript referring to this supplementary section and Figs. S5-9. The new text (lines 435:437) reads:*

See Sect.1 and Figs. S5-9 in the Supplement about the evolution of the aragonite saturation horizon ($\Omega AR$=1, ASH) in the RCP 4.5 control run and the A2 and A3 experiments.

3) What is the role of land carbon cycle here? How much of the outgassed $CO_2$ can be attributed to (or compensated by) the terrestrial $CO_2$ flux? Can the land carbon cycle feedback be quantified by turning on/off the UVic land carbon component?

*Thank you for this interesting point. The response of the land carbon cycle due to an injection-related atmospheric carbon reduction is mainly governed by the reduced $CO_2$ fertilization effect on net primary productivity and the temperature related decrease in heterotrophic soil respiration, which is something that we investigated in Reith et al., 2016. However, feedbacks from the terrestrial system to changes in atmospheric $CO_2$ are among the largest uncertainties (e.g., Schimel et al., 2015). In order to attribute the terrestrial carbon cycle response to the outgassed amount of injected $CO_2$, we would need to perform partially coupled simulations, i.e., simulations with only the carbon cycle model components experiencing rising $CO_2$ (bioge-ochemically (BGC) coupled) and only the radiation model components experiencing rising $CO_2$ (radiatively (RAD) coupled) (e.g. Schwinger et al., 2014). Not having these idealized model experiments, it can be stated that the difference between ex-periments A1 an A2 (a net loss of carbon from land in the successful (climate target) A2 experiment of about 100 Gt C, Fig. R6) is part of the additional carbon injected in experiment A2 compared to experiment A1.*

[Figure]

**Figure R6:** Cumulative land fluxes (Gt C) for the A1 simulation (black line) and the A2 simulation (red line).

*No, the land carbon cycle feedbacks cannot easily be quantified by turning on/off the land component of the UVic model. This is in particular due to the fact that the applied RCP 4.5 $CO_2$ emission scenario includes historical land-use carbon emissions (line 114), which would lead to a different climate response if it was used when the land component was switched off. Fur-ther, in a coupled system like the UVic model turning off a single component will affect the remaining system. Thus, deter-mining the terrestrial role by such additional experiments would require a noLand model spinup and modification of the RCP 4.5 emission scenario so that it reproduces a climate response consistent with the response to the full RCP 4.5 emissions in the fully coupled model. Thereafter, one could repeat the model runs with this noLand version. Then, to disentangle and quantify the terrestrial processes involved, we would further need to separately perform additional partially coupled runs (only biogeochemically coupled and only radiatively coupled simulations). As the terrestrial biosphere has little influence on the cumulative $CO_2$ injections (Reith et al., 2016), we are convinced that we would not gain any important insights from such time-consuming experiments.*

*We added a new subsection (3.1.2 Land response) to the manuscript in order to better address this issue. The new text (lines 285:297) reads:*

*"Biological processes primarily control the exchange of atmospheric $CO_2$ with the land, where the majority of the carbon is stored in soils and permafrost. $CO_2$ is removed from the atmosphere by plant photosynthesis and primarily returned to the atmosphere by respiration and other processes such as fire (Cias et al., 2013). As long as primary production (GPP; i.e. gross photosynthetic carbon fixation) is greater than carbon losses due to respiration and processes such as fire, the land will be a carbon sink (Le Quere et al., 2016). If this balance changes then the land can become a source of $CO_2$ to the atmosphere. In our simulations as emissions increase and decrease (with oceanic CCS), eventually reaching net negative- and/or zero- emission, the terrestrial carbon cycle is perturbed and can switch from a sink to a source. The magnitude of the land carbon cycle response due to an injection-related atmospheric carbon reduction, and eventual increase as in the A1 experiment, (see Section 3.1) is mainly governed by the reduced $CO_2$ fertilization effect on net primary productivity and the temperature related change in heterotrophic soil respiration, responses that have been investigated in Reith et al., 2016. In all simulations the land changes from a carbon sink to source, eventually reaching almost a balance (almost zero net flux) in the A2 and A3 simulations. In the A1 simulation, the increase in atmospheric $CO_2$ after a period of decline (Fig. 1 c) is not able to overcome the temperature effect that elevates respiration rates and the land continues to perennially lose carbon."*

[Figure]

**Figure S5:** Depth of aragonite saturation horizon (ASH) at the beginning of the RCP 4.5 control run (year 2020).

[Figure]

**Figure S6:** Relative changes in depth of aragonite saturation horizon (ASH) in the RCP 4.5 control run in 2100, i.e., year 2100 minus year 2020.

[Figure]

**Figure S7:** Relative changes in depth of aragonite saturation horizon (ASH) between the A2 simulation (year 2100) and the RCP 4.5 control run (year 2020).

[Figure]

**Figure S8:** Relative changes in depth of aragonite saturation horizon (ASH) in the RCP 4.5 control run in 3020, i.e., year 3020 minus year 2020.

[Figure]

**Figure S9:** Relative changes in depth of aragonite saturation horizon (ASH) between the A2 simulation (year 3020) and the RCP 4.5 control run (year 2020).